# Sodium Butyrate Effectiveness in Children and Adolescents with Newly Diagnosed Inflammatory Bowel Diseases—Randomized Placebo-Controlled Multicenter Trial

**DOI:** 10.3390/nu14163283

**Published:** 2022-08-11

**Authors:** Anna Pietrzak, Marcin Banasiuk, Mariusz Szczepanik, Agnieszka Borys-Iwanicka, Tomasz Pytrus, Jarosław Walkowiak, Aleksandra Banaszkiewicz

**Affiliations:** 1II Gastroenterology Department, Centre of Postgraduate Medical Education, 01-813 Warsaw, Poland; 2Gastroenterology Department, Bielanski Hospital, Ceglowska 80 Str., 01-807 Warsaw, Poland; 3Department of Pediatric Gastroenterology and Nutrition, Medical University of Warsaw, 02-091 Warsaw, Poland; 4Department of Pediatric Gastroenterology and Metabolic Diseases, University of Medical Sciences, 60-572 Poznan, Poland; 5Department of Paediatrics, Gastroenterology and Nutrition, Wroclaw Medical University, 50-369 Wroclaw, Poland

**Keywords:** Crohn’s disease, ulcerative colitis, sodium butyrate

## Abstract

Background: Butyric acid’s effectiveness has not yet been assessed in the pediatric inflammatory bowel disease (IBD) population. This study aimed to evaluate the effectiveness of oral sodium butyrate as an add-on to standard therapy in children and adolescents with newly diagnosed IBD. Methods: This was a prospective, randomized, placebo-controlled multicenter study. Patients aged 6–18 years with colonic Crohn’s disease or ulcerative colitis, who received standard therapy depending on the disease’s severity, were randomized to receive 150 mg sodium butyrate twice a day (group A) or placebo (group B). The primary outcome was the difference in disease activity and fecal calprotectin concentration between the two study groups measured at 12 weeks of the study. Results: In total, 72 patients with initially active disease completed the study, 29 patients in group A and 43 in group B. At week 12 of the study, the majority of patients achieved remission. No difference in remission rate or median disease activity was found between the two groups (*p* = 0.37 and 0.31, respectively). None of the patients reported adverse events. Conclusions: A 12-week supplementation with sodium butyrate, as adjunctive therapy, did not show efficacy in newly diagnosed children and adolescents with IBD.

## 1. Introduction

Inflammatory bowel disease (IBD): Crohn’s disease (CD) and ulcerative colitis (UC) are chronic gastrointestinal disorders with periods of exacerbation and remission. The disease develops as a result of an abnormal immune response in the gastrointestinal mucosa in genetically predisposed individuals exposed to certain environmental conditions. In recent years, more and more data support the interpretation that gastrointestinal dysbiosis has a huge role in the pathogenesis of IBD [1,2].

Butyric acid, a short-chain fatty acid (SCFA), has a number of properties that may affect various diseases of the gastrointestinal tract [3,4]. Butyric acid is an energy material for normal intestinal epithelial cells and has a trophic effect on the normal intestinal mucosa [5]. Additionally, it shapes the intestinal microbiota by stimulating the growth of saprophytic flora such as *Lactobacillus rhamnosus*. It inhibits the growth of other pathogens, such as *Escherichia coli* [6]. The appropriate concentration of SCFA, including butyric acid, helps to maintain the correct pH in the intestinal lumen, which is another mechanism that protects against the invasion of microorganisms. SCFAs also have an immunomodulatory effect, e.g., they play a beneficial role in the metabolism of adipose tissue, and their concentration in the intestinal lumen correlates with bodyweight [7].

Butyric acid is commonly present in the human diet, but the amounts contained therein are so small that they cannot affect the epithelium of the small intestine. The richest sources of butyric acid in the large intestine are resistant to digestion starch, oats and wheat bran. Eating foods such as partially ground grains, seeds, and vegetables provides resistant starch to bacteria that produce butyric acid [8]. The main producers of butyric acid are microorganisms living in the large intestine, mainly sugar-fermenting bacteria, such as *Clostridioides* spp., *Eubacterium* spp., *Fusobacterium* spp., and others [9].

In experimental studies, an anti-inflammatory effect of butyric acid in IBD was found [10,11,12]. Oral butyric acid supplementation has been shown to reduce dysbiosis in UC patients [13]. On the other hand, there are not many clinical trials assessing the influence of butyric acid on IBD activity, and their results are divergent. All were conducted among adults. So far, the effectiveness of oral butyric acid supplementation in children with IBD has not been assessed, which was the purpose of this study.

## 2. Materials and Methods

The study was prospective, randomized, and placebo-controlled. It was carried out in 3 pediatric clinical centers for the diagnosis and treatment of pediatric IBD in Warsaw, Wroclaw and Poznan.

Children who met the following inclusion criteria were enrolled in the study: 6 to 18 years of age; newly diagnosed, based on the modified Porto criteria, with IBD with colon involvement; informed consent of the child’s parents or guardians to participate in the study. Exclusion criteria from the study included: age < 6 years; taking probiotics or dietary supplements in the last 2 weeks prior to study enrollment; lack of consent of parents or guardians to participate in the study. Only children with a normal nutritional status (dominant BMI = 19.6), without nutritional deficiencies in the laboratory assessment and with a dietitian-guided IBD-directed diet were included. No other dietary intervention, including supplements, was provided.

Sodium butyrate or placebo was administered orally in capsule form every 12 h for 12 weeks. All patients during the study received standard treatment depending on the type of disease (CD or UC) and degree of activity, while they received no other dietary supplements, including probiotics and prebiotics.

Patients were randomized on the basis of a computer-generated randomization list and were assigned to one of two groups: group A received butyric acid at a dose of 150 mg (Debutir^®^, Poland), and group B received 150 mg placebo. The verum and placebo capsules had the same shape, color and size. The capsules were packed in 60 pieces in matching boxes; the only difference was labeling with the letter A or B. The randomization list was overseen by a person not related to the study. The researcher assessing the severity of disease activity after the 12 weeks of the study was blinded.

When it was possible, a stool specimen was collected from the patient prior to study entry and 12 weeks after study entry to assess calprotectin concentration.

The Pediatric Crohn’s Disease Activity Index (PCDAI) was used to assess CD activity, and the Pediatric Ulcerative Colitis Activity Index (PUCAI) to assess UC activity [14,15]. Remission was defined as less than 10 points for both PCDAI and PUCAI. The Paris criteria were used to assess the extent of the disease.

The primary outcome measure was the difference in remission rate and disease activity between the two study groups assessed by PCDAI and PUCAI scores measured at 12 weeks. The secondary endpoints included: (1) the difference in disease activity between the two study groups measured at 12 weeks, assessed separately for CD and UC patients; (2) the difference in calprotectin concentration between the time of study entry and 12 weeks thereafter; (3) side effects. We also performed sub-analyses assessing potential differences depending on gender, age (6–12 and 13–18 years old), disease extent and activity, and the therapy used. Thanks to similar numerical cut-off levels for disease severity (definition of remission, mild, moderate and severe disease in PCDAI and PUCAI scales), we could assess both scales together.

The sample size was established based on previous research that showed the effectiveness of butyrate supplementation in adult patients (53% vs. 6.25%) [16]. Assuming the 80% power of the test and a significance level = 5%, each group should count 36 patients, that is 72 patients overall. Estimating the non-compliance rate of 10%, the final sample size was calculated for 79 patients.

Numeric data were collected in Excel spreadsheet tables. A survey conducted with yes/no questions was translated into the tables by assigning values based on false, no = 0 and true, yes = 1. After checking the normality of distribution, the results were analyzed using analysis of variance. In the absence of normal distribution, non-parametric tests with appropriate posthoc tests were used. For the comparisons between the groups, test chi2 (significance level (*p*-value) 0.01), Fisher exact test (significance level (*p*-value) 0.01), Spearman’s rank correlation, and Pairwise Wilcoxon Rank Sum Tests with Bonferroni correction were used depending on group size. The analysis was performed using GraphPad Prism v.5.02. The differences between groups are presented as mean +/− SD using bar plots. A value of *p* < 0.05 indicates that the differences were statistically significant, while a *p* values in the range 0.1–0.05 considers trends.

The study was conducted according to the CONSORT statement for randomized controlled trials.

Written informed consent was obtained from all the guardians of the subjects involved in the study. The study was conducted according to the guidelines of the Declaration of Helsinki and approved by the Institutional Review Board of the Medical University of Warsaw (protocol code 41/2013, date of 30 January 2013). The study was registered in the ClinicalTrials.gov database (registration number NCT05456763). The study was performed using the statutory funds of the university. Article-processing charges were covered by the Centre of Postgraduate Medical Education.

## 3. Results

Figure 1 presents the flowchart of the study.

One hundred consecutive children with newly diagnosed IBD referred to the research centers were assessed for eligibility. Eighteen were in remission, one received probiotics, and one withdrew her consent before enrollment. Patients were excluded from the study who did not meet inclusion criteria. In total, 80 patients were included into the study (per-protocol analysis), 35 in the A group and 45 in the B group. The data of nine patients concerning treatment results were incomplete, and they were excluded from further evaluation. In total, 72 patients with a median age of 13.5 years (including 42 patients with CD, 60% boys) were analyzed (per protocol analysis). CD and UC patients were comparable in terms of disease activity, calprotectin levels and the percentage of patients treated with amino-salicylates, steroids and antibiotics. Patients with CD received thiopurines and anti-TNF statistically more often. The disease was mild in 24 CD (57.0%) and 11 UC patients (36.6%). Severe disease was diagnosed in seven patients in both groups (16.6 and 23.3%, respectively). Patients with colonic involvement, defined as E1/E2/E3/E4 for UC and L2/L3 for CD (regardless of the involvement of other sections of the GI tract, the CD form and the age of the child), comprised the majority of the study group.

Groups A, receiving the studied drug, and B, receiving placebo, were comparable in terms of the type of the disease, age, gender, disease activity, calprotectin level and received treatment. The baseline characteristics are presented in Table 1.

At week 12, the majority of patients in groups A and B achieved remission (68%). There was no difference in remission rate between groups (A 62%, B 72%; *p* = 0.371). No difference in median disease activity was found between groups A and B: 10 points (SD = 14.3) vs. 5 points (SD = 9.53), *p* = 0.08. There was also no difference between the median calprotectin concentration: 450 µg/g (SD = 667) vs. 585 (SD = 707), *p* = 0.466 (the calculation was performed only for complete input and output data).

After the study period, the remission rates achieved were 62% in CD patients and 76% in UC patients, regardless of received intervention (details, Figure 2). The endpoint PCDAI and PUCAI scores did not differ between groups.

We did not find any differences between groups concerning gender, age, disease activity, and therapy used between the two groups. We also performed subgroup analyses assessing age younger or older than 13 years, and multivariate analyses addressing non-severe (mild or moderate) vs. severe disease, as well as aggressive immunosuppressive treatment (antibiotics, steroids and biologics in different combinations), and we did not find any differences between butyrate and placebo effectiveness (details, Table 2).

None of the patients in the study reported adverse events.

## 4. Discussion

The results of our study, which was the first in a pediatric population, found supplementation with sodium butyrate to be ineffective in the add-on treatment of newly diagnosed children and adolescents with IBD.

To date, only a few studies have investigated oral sodium butyrate supplementation in adult IBD patients, and the results are inconsistent. Our results are in line with those of the Italian study, albeit conducted in a smaller number of adult patients (*n* = 49) with CD and UC [16]. In Facchin et al.’s randomized, controlled trial, after 2 months of oral administration of 1800 mg of sodium butyrate or a placebo per day in addition to conventional treatment, no changes in IBD activity or calprotectin levels were observed. As in our group, in the Italian study, patients with CD predominated. All patients were diagnosed at least six months before the start of the study and therefore had prior treatment, including surgical treatment (although major surgery was an exclusion criterion from the study). Additionally, the activity of the disease could have been influenced by environmental factors not present in our group of patients, such as smoking or taking prebiotics.

In contrast, in the Vernia et al. study, 30 UC patients treated with fixed doses of mesalazine were randomized: one group received 4 g of sodium butyrate for 6 weeks, and the other group received a placebo [17]. After completion of the intervention, 7/15 patients receiving sodium butyrate achieved remission of their disease and a further 4/15 improved, while patients receiving placebo showed 5/15 remission, and 5/15 patients improved. In an observational study by Di Sabatino et al., 13 patients with CD were studied. After 8 weeks of oral supplementation of 2 × 2 g sodium butyrate, disease remission of 53% (7/13) was achieved in patients with CD, and a further 2/13 achieved a reduction in disease activity [18]. All patients had mild to moderate disease activity and were receiving mesalazine. A multicenter observational study by Assisi et al. showed that the administration of 900 mg of sodium butyrate supplemented with an additional 750 mg of inulin is effective in reducing UC activity [19].

Our results are consistent with the results of a recently published systematic review of studies assessing the effectiveness of sodium butyrate enemas in the treatment of adult UC patients [20]. Of the eight studies included in the review, only one decrease in disease activity was noted.

The reason for the lack of efficacy of sodium butyrate in our study is unclear. One potential explanation relates to dose, as a clear target concentration in the distal gut has not been established. Butyric acid, along with other SCFAs produced by the bacterial fermentation of unabsorbed carbohydrates, is an important colonocyte nutrient that can reach millimolar concentrations in the colons of healthy individuals [21]. However, when given orally, if not otherwise prepared, SCFA are largely absorbed in the proximal small bowel and metabolized similarly to dietary fatty acids, thus limiting colonic bioavailability [21]. There is growing evidence that the formulation of butyrate using pH-sensitive encapsulation technologies, such as the one used in our study, can significantly delay enteric release, reduce small intestinal absorption and enhance colonic delivery [21,22,23,24]. However, our results, consistent with those of others, indicate that further work is needed to establish whether effective doses can be achieved with existing formulations in this patient population.

The appropriate duration of supplementation use is also not clearly established. In some studies, 4–8 weeks of sodium butyrate enema treatment in UC patients was sufficient to see improvement [25]. However, it cannot be ruled out that effective butyrate supplementation should be much longer than the 12 weeks we used in our study. The divergent results obtained by different authors may be explained by the fact that sodium butyrate works more effectively in patients with lower disease activity or in remission (then it can be said to help maintain remission). A patient with exacerbated IBD usually has diarrhea, sometimes very severe, which on the one hand reduces the amount of butyrate retained in the gastrointestinal lumen, and on the other hand, can significantly reduce the number of bacteria affected by butyrate. This hypothesis is confirmed in the results of the study by Facchin et al., who after oral supplementation with sodium butyrate found changes in the microbiota composition only in patients in remission, and did not find any changes in the microbiota composition in patients with exacerbation of the disease [16]. This was also confirmed in a recently published study by Vernero et al. The authors observed UC patients in disease remission treated with mesalazine alone—some patients received daily oral doses of 2 × 500 mg sodium butyrate, while the others did not receive supplementation [26]. At 12 months, 83.3% (15/18) of patients receiving sodium butyrate and 47.6% (10/21) of patients not receiving sodium butyrate were still in remission, *p* = 0.022.

Sodium butyrate is very safe when used in patients with IBD. During the course of the study, no patients reported adverse events. Worth emphasizing is the high safety of sodium butyrate reported in all published studies, both in studies where high doses of butyrate were given orally, e.g., 4 g/day for 8 weeks, such as in the study by Di Sabatino et al., and in studies where sodium butyrate was administered rectally [18].

It is difficult to summarize the already-published studies assessing the effect of sodium butyrate supplementation on IBD activity. This is due to their high heterogeneity—the use of different doses, confection methods, forms and routes of administration (oral, rectal) of sodium butyrate, different durations of supplementation, the use of different endpoints, etc. This is also due to the high heterogeneity of IBD patients. The disease—in fact, the two diseases, CD and UC—have different localizations of inflammatory lesions in the gastrointestinal tract, and patients are administered various types of treatment. This great heterogeneity may explain why sodium butyrate is so effective in treating inflammatory lesions in the intestines of a mouse model, and is so ineffective in the treatment of IBD in humans [27,28,29]. The second explanation for the ineffectiveness of sodium butyrate in the treatment of IBD is the result of a study by Vancamelbeke et al. published in 2019, who created an ex vivo model of the epithelial cell layer of the gastrointestinal tract of UC patients to assess the effect of sodium butyrate on the protective effect of the epithelium. In this model, they found that butyrate not only does not prevent the negative effects of pro-inflammatory cytokines such as TNF alpha or IFN gamma, but it actually increases their pro-inflammatory effects [30]. Undoubtedly, further extensive experimental research is necessary to elucidate the true role of sodium butyrate in the lumen of the gut.

Our study is the first in the IBD pediatric population, and it was conducted to evaluate the effectiveness of oral sodium butyrate on disease activity. The limitations of our study include the relatively small size of the study group; however, it is still one of the largest studies on IBD patients. We did not assess the effect of sodium butyrate supplementation on the composition of the intestinal microbiota, but our goal was to achieve a clinical evaluation and practical application of the obtained results. Currently, in medical practice (unfortunately), intestinal microbiota testing is not routinely performed either for the diagnosis of IBD or the assessment of treatment. In light of the recently published review summarizing the doses of butyrate used in experimental and clinical trials, it seems that we might have used a too-low dose of sodium butyrate [31]. However, it is still the recommended dose; the lack of studies in children and the small number of studies on oral supplementation made us careful in choosing a dose size. The sodium butyrate preparation was well tolerated by the patients at the tested dose.

## 5. Conclusions

In our study, we did not demonstrate the effectiveness of 12 weeks of oral sodium butyrate at a dose of 300 mg/day on the disease activity of IBD in children. It is worth noting that not just the oral dose, but also the capsule formulation, is important to consider when making conclusions about sodium butyrate’s efficacy in our cohort. However, we believe that the results of our study will contribute to further studies that will determine which patients with IBD may benefit from sodium butyrate supplementation. Further clinical trials on large groups of patients are needed to establish if IBD patients may benefit from sodium butyrate.

## Figures and Tables

**Figure 1 nutrients-14-03283-f001:**
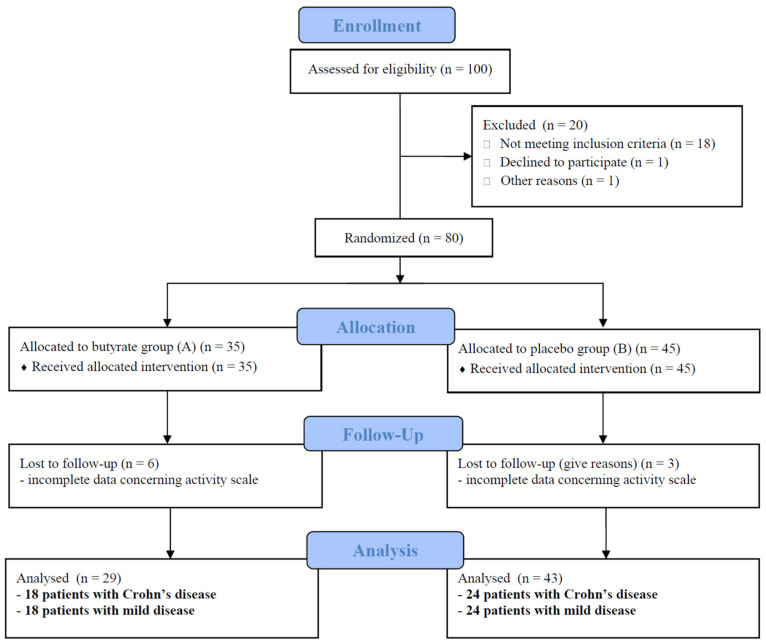
Flowchart of the study according to the CONSORT statement.

**Figure 2 nutrients-14-03283-f002:**
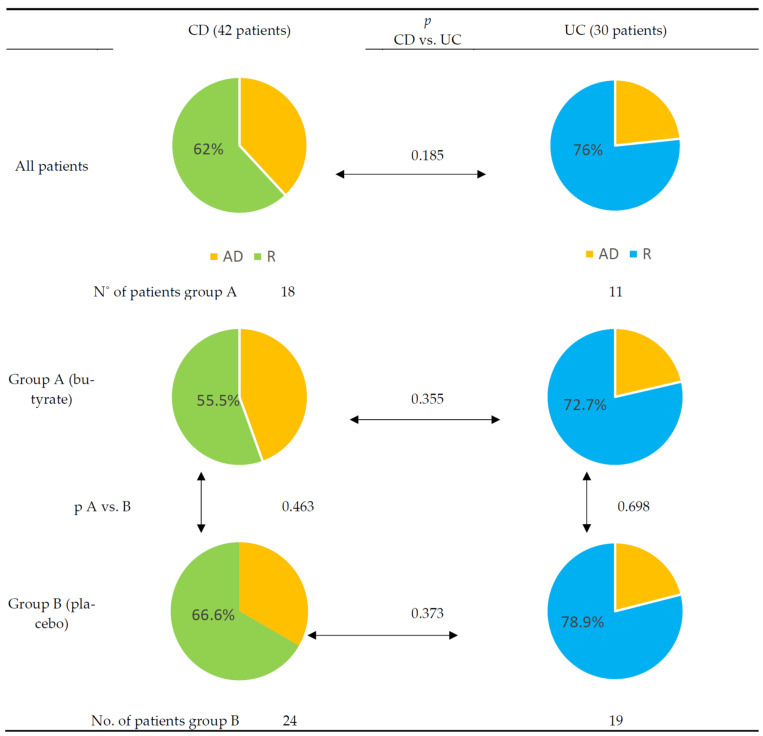
Differences between Crohn’s disease and ulcerative colitis cohorts after 12 weeks of treatment. Yellow—active disease (AD). Green—remission (R) in Crohn’s disease (CD). Blue—remission (R) in ulcerative colitis (UC).

**Table 1 nutrients-14-03283-t001:** Baseline characteristics of the study group.

Number of Patients		Study Group 72	Group A (Butyrate) 29	Group B (Placebo) 43	*p*
Crohn’s disease (No. of patients)		42 (58.3%)	18 (62.1%)	24 (55.8%)	0.597
Mean age (years)		13.3 (6–18)	12.9 (6–17)	13.5 (7–18)	0.6
Gender (male)		43 (59.7%)	14 (48.3%)	29 (67.1%)	0.104
Median weight (kg)		46 (SD 14.8)	45 (SD 15.3)	46.5 (SD 14.2)	0.329
Median height (cm)		158 (SD 19.4)	158 (SD 20.9)	159 (SD 17.9)	0.620
Baseline activity index score (points)	Mean	34.5 (12.5–85)	34.7 (12.5–75)	34.3 (12.5–85)	0.554
	Median	31.5	30	32.5	-
Calprotectin level (points)	Mean	1111.1 (50–1800)	1090 (50–1800)	1122.2 (50–1800)	0.599
	Median	1100	900	1300	-
Amino-salicylates		70 (97.2%)	28 (96.5%)	42 (97.7%)	0.776
Steroids		8 (11.0%)	3 (10.3%)	5 (11.6%)	0.865
Thiopurines		50 (69.5%)	21 (72.4%)	29 (67.4%)	0.653
Anti-TNF		10 (13.9%)	4 (13.8%)	6 (13.9%)	0.985
Antibiotics		18 (25.0%)	8 (27.6%)	10 (23.3%)	0.677
Disesae extention (Paris criteria) *—colon involvement (N^o^ of patients)					
UC: E1/E2/E3/E4		5/7/3/15	3/3/1/8	2/4/2/7	n.s.
CD: L2/L3		5/34	3/14	2/20	n.s.

* Colonic involvement according to Paris criteria: UC—E1: ulcerative proctitis, E2: left-sided UC (distal to splenic flexure), E3: extensive (hepatic flexure distally), E4: pancolitis (proximal to hepatic flexure); CD—L2: colonic, L3: ileocolonic. TNF—tumour necrosis factor, n.s.—not significant.

**Table 2 nutrients-14-03283-t002:** Comparison of butyrate effectiveness as an add-on therapy in achieving remission in different subgroups.

Number of Patients		Study Group 72	Group A 29	Group B 43	*p*
Mild disease (≤10 points)	All	42	18	24	-
	Rem.	25 (59.5%)	9 (50%)	16 (66.7%)	0.276
Moderate/severe disease (>10 points)	All	30	11	19	-
	Rem.	24 (80%)	9 (81.8%)	15 (78.9%)	0.85
Excluding patients on steroids	All	64	26	38	-
	Rem.	31 48.4%)	15 (57.7%)	24 (63.2%)	0.38
Excluding patients on antibiotics	All	54	21	33	-
	Rem.	40 (70.1%)	14 (66.7%)	26 (78.8%)	0.322
Excluding patients on steroids, antibiotics and anti-TNF	All	42	15	27	-
	Rem.	29 (69.0%)	9 (60.0%)	20 (74.1%)	0.344
Patients between 6 and 12 years old	All	27	14	13	-
	Rem.	19 (70.4%)	9 (64.3%)	10 (76.9%)	0.302
Patients between 13 and 18 years old	All	45	15	30	-
	Rem.	30 (66.7%)	9 (60.0%0	21 (70.0%)	0.502

Rem—remission. Assessment after 12 weeks of treatment. Grey rows shows number of patients who achieved remission in every sub-group analyzed, shown as percentage in brackets.

## Data Availability

All data supporting reported results are digitalized and can be found in the submission’s guarantor office.

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
