# Peer review of "Sodium Butyrate Effectiveness in Children and Adolescents with Newly Diagnosed Inflammatory Bowel Diseases—Randomized Placebo-Controlled Multicenter Trial"

_nutrients, 2022, doi:10.3390/nu14163283_

Round 1

Reviewer 1 Report

The article by Pietrzak et al. entitled: Sodium butyrate effectiveness in children and adolescents with 2 newly diagnosed inflammatory bowel diseases – randomized 3 placebo-controlled multicentre trial showed for the first time the effect of Butyric acid’s in the pediatric inflammatory bowel disease population. This prospective, randomized, placebo-controlled multicentre study is well designed and the results are detailed described. I have no comment for the authors, in my opinion the article could be accepted in the present form.

Author Response

We would like to thank you for your valuable opinion and assure you that we will make every effort that the published work meets the reviewers' expectations.

Reviewer 2 Report

This is a well-written study, investigating the role of oral short-chain fatty acid (SCFA) supplementation in pediatric IBD patients.

1. While this reviewer applauds the investigators for conducting a well-designed protocol, the negative results are not at all surprising. Butyric acid, along with other SCFA species, is an important colonocyte nutrient, and provides significant energy salvage from the bacterial metabolism of unabsorbed carbohydrates. When give orally, SCFAs are absorbed in the proximal small bowel, and metabolized similarly to other fatty acids. Accordingly, the trivial amount of additional calories provided in this study would certainly not be expected to improve overall nutritional status. Furthermore, considering sodium butyrates proximal absorption and metabolism, direct effects on enterocyte kinetics or IBD-associated inflammation could be predicted as minimal.

2. As stated above, the negatived findings herein are not at all surprising. However, considering the nutritional focus of this study, information regarding the nutritional status and nutrient intake of study subjects should have been provided.

Author Response

We would like to thank you for your valuable opinion and assure you that we will make every effort that the published work meets the reviewers' expectations. We agree that considering the physiology of SCFA the results are not surprising; we did not expect a spectacular effect of butyrate supplementation. However, in diseases such as IBD, where all the factors influencing the course of the disease and treatment outcomes are unknown, any intervention with a chance of even a slight improvement in treatment is worth checking. Several scientists, following similar reasoning, conducted research with the sodium butyrate among adult patients with IBD. Moreover, thanks to the results of research such as ours, we can more consciously advise parents of children with IBD to take (or not to take) supplements.

According to the Reviewers’s suggestion we included data on nutritional status of patients to the Table 1. We also added the proper paragraph into the Methods section.

We added those data to the table 1 for clarifying.

Description

Study group

Sodium butyrate

Placebo

P

Weight

Mean

44,2

43,2

45,1

0,329

Median

46

45

46,5

SD

14,8

15,3

14,2

Height

Mean

155,4

154,3

156,5

0,620

Median

158

158

159

SD

19,4

20,9

17,9

 No other dietary intervention, including supplements, was provided.

The manuscript was edited for proper English language, grammar, punctuation, spelling, and overall style by one or more of the highly qualified native English speaking editors at Cambridge Proofreading.

Reviewer 3 Report

In this prospective double blind randomized clinical trial Pietrzak et al demonstrated that add-on treatment of sodium butyrate in children with inflammatory bowel disease (IBD) does not bring any benefit in achieving clinical remission at 12 weeks. Main comments:

1) Page 2 lines 86-89: such sentences should be supported by references.

2) Page 3 line 110: what does “SI 0.01” mean?

3) Table 1: even if not statistically significant, please report each p value.

4) Authors affirmed that 18 patients were in remission at baseline, while at 12 weeks 68% were in remission. Does 68% includes patients who maintained remission? I believe it would be more useful to differentiate the statistical analysis between those who maintained remission and those achieving remission.

5) A sub-group of effectiveness in CD and UC may be useful.

6) Table 2 is hard to understand. Please check numbers in group B (430?).

7)  In table 1, please report as well disease extension according to Paris criteria.

Author Response

We would like to thank you for your valuable opinion and assure you that we will make every effort that the published work meets the reviewers' expectations. Answering to the reviewer comments:

1) We supported the Pediatric Activity Indexes (lines 86-89) by references.

2) In statistical methods (line 110), Sl means significance level, which implies the p-value we decided to use in statistical analysis as statistically significant. As the reviewer pointed the second letter “l” can be easily confused with the capital “i”. Thus we clarified the terms.

3) Table 1: we added a column with p-value. Due to formatting difficulties we marked this column in red.

4) Patients in remission weren’t assessed in PP analysis. We primarily assessed 100 patients from whom 20 were excluded (among them those 18 in remission). We lost to follow-up next 9 children, thus we analyzed 72 children with active disease at the beginning. From them, 68% achieved remission in 12 weeks. We clarified the description in materials and method section.

5) We assessed UC and CD sub-groups (results section, lines 160-162) and we didn’t find any differences. For easier interpretation for the reader we added figure 1 presenting sub-groups comparison in graphs.

6) Numbers are checked and corrected. We accessory described rows below the table and added percentage values for better interpretation.

7) Table one – disease extension according to Paris criteria added to the table. Also description of the group added in result section (lines: 142-144).

Please find attached Figure 1 and additional calculations for disease extension according to Paris with calculation as word file. 

We hope that results are described more precise in the present form. The manuscript was edited for proper English language, grammar, punctuation, spelling, and overall style by one or more of the highly qualified native English speaking editors at Cambridge Proofreading.

Round 2

Reviewer 2 Report

See prior comments

Author Response

We would like to thank you for your important and valuable comments. We provided baseline general nutritional data such as weight and height of the participants and BMI with calculated median, mean and dominant values. We added information about laboratory findings and general dietary recommendations to the material and methods section of the manuscript.

Thank you for your suggestion to include nutritional issues in our study. Unfortunately, following the example of other authors, we did not take it into account, when planning a study to assess the effectiveness of butyrate in the treatment of children with IBD. However, we will certainly include nutritional aspects such as dietary butyric acid content in our future research on sodium butyrate.

We now also added information about our expectations regarding butyrate in the discussion section (lines 217-234). We didn't expect overall nutritional status improvement although, we admit, that we did not check it. Maybe because it has never been included in any studies before. Considering available data and previous studies' results, we assumed that the crucial function of SCFAs would be the colonocytes regeneration and restoration with mucosal healing.

Thank you for paying attention to the role of butyrate exposure in different parts of the human intestine. We do agree with the Reviewer 2 and  the Editor that colonic exposure to butyrate is more important than oral dose per se. There is growing evidence that there is a unique formulation of the butyrate-containing capsules which deliver the substance to the colon. According to those studies, microencapsulated butyrate is formulated as a colon-release preparation. In our formulation, a pH-dependent polymer with anionic methacrylate copolymer was used, and it had been validated for targeted colon delivery. From a pharmacokinetic perspective, having evidence that the butyrate administered orally is delivered mainly to the colon rather than the proximal small intestine is essential, especially given the existing discussion around the choice and suitability of dose. We hope that the paragraph in its new form will meet with Reviewer’s approval.

paragraph added to the discussion

Although butyric acid, along with other SCFAs, is an important colonocyte nutrient and provides significant energy salvage from the bacterial metabolism of unabsorbed carbohydrates, when given orally, if not otherwise prepared, are absorbed in the proximal small bowel and metabolized similarly to other fatty acids. Thus direct effects on enterocyte kinetics or IBD-associated inflammation could be predicted as minimal. However, there is grooving evidence that the butyrate-containing capsules have a unique formulation that delivers the substance to the colon [21 - Spina et al.]. According to those studies, microencapsulated butyrate is formulated as a colon-release preparation using a pH-dependent polymer with an anionic methacrylate copolymer. It had been validated for targeted colon delivery [22 – Cole et al.]. In our study, we used patented microencapsulated preparation, providing the highest colonic butyrate concentration. Comparative analysis of different confection modalities proved microencapsulation to be the most effective. It was comparable with several times higher doses of nonencapsulated butyrate [23 – Banasiewicz et al., 24 - Roda et al.]. From a pharmacokinetic perspective, having evidence that the butyrate administered orally is delivered mainly to the colon rather than the proximal small intestine is essential, especially given the concerns about the choice and suitability of dose. With respect to the hypothesis being tested, in our opinion, colonic exposure is more important than oral dose per se.

Bibliography

  1. Spina L., Cavallaro F., Fardowza N.I. et al. Butyric acid: pharmacological aspects and routes of administration. Dig Liver Dis 2007; S1:7-11.
  2. Cole E.T., Scott R.A., Connor A.L. et al. Enteric coated HPMC capsules designed to achieve intestinal targeting. Int J Pharm 2002; 231:83-95.
  3. Banasiewicz T., Borycka-Kiciak K., Kiciak A. Porównanie profilów uwalniania maÅ›lanu sodu w jelicie dla produktów dostÄ™pnych na polskim rynku (ang. Comparison of release profiles sodium butyrate in the intestine for products available on the Polish market). Farmacja Praktyczna 2019/2020; 12/1(119):40-42.
  4. Roda A., Simoni P., Magliulo M. et al. A new oral formulation for the release of sodium butyrate in the ileo-cecal region and colon. World J Gastroenterol 2007; 21:1079-1084.

Please also find attached comparison of sodium butyrate release place in the digestive tract (reprinted with permission from Banasiewicz et al, Farmacja Praktyczna 2019).

Article in Polish. Brand names pointed below bars. Translating from Polish: żołądek = stomach; dwunastnica, jelito czcze = duodenum, jejunum; jelito kręte = ileum; jelito grube = colon.

Reviewer 3 Report

Answers were satisfactory

Author Response

(The authors gave the same response as above.)
